# Genome-Wide Analysis of the Cis-Prenyltransferase (CPT) Gene Family in *Taraxacum kok-saghyz* Provides Insights into Its Expression Patterns in Response to Hormonal Treatments

**DOI:** 10.3390/plants14030386

**Published:** 2025-01-27

**Authors:** Liyu Zhang, Huan He, Jiayin Wang, Pingping Du, Lili Wang, Guangzhi Jiang, Lele Liu, Lu Yang, Xiang Jin, Hongbin Li, Quanliang Xie

**Affiliations:** 1Key Laboratory of Xinjiang Phytomedicine Resource and Utilization of Ministry of Education, College of Life Sciences, Shihezi University, Shihezi 832003, China; zhangliyu981125@163.com (L.Z.); he_huan026@163.com (H.H.); wjyinee@163.com (J.W.); dopingping@126.com (P.D.); lfyroquz@163.com (L.W.); csghzyl@foxmail.com (G.J.); lzjl6774@163.com (L.L.); luyang1230518@163.com (L.Y.); 2Ministry of Education Key Laboratory for Ecology of Tropical Islands, Key Laboratory of Tropical Animal and Plant Ecology of Hainan Province, College of Life Sciences, Hainan Normal University, Haikou 571158, China

**Keywords:** *Taraxacum kok-saghyz*, TkCPT/CPTLs, natural rubber, gene expression, ethylene, methyl jasmonate

## Abstract

*Taraxacum kok-saghyz* (TKS) is a natural rubber (NR)-producing plant with great development prospects. Accurately understanding the molecular mechanism of natural rubber biosynthesis is of great significance. Cis-prenyltransferase (CPT) and cis-prenyltransferase-like (CPTL) proteins catalyze the elongation of natural rubber molecular chains and play an essential role in rubber biosynthesis. In this study, we performed a genome-wide identification of the *TkCPT*/*CPTL* family, with eight *CPT* and two *CPTL* members. We analyzed the gene structures, evolutionary relationships and expression patterns, revealing five highly conserved structural domains. Based on systematic evolutionary analysis, CPT/CPTL can be divided into six subclades, among which the family members are most closely related to the orthologous species *Taraxacum mongolicum.* Collinearity analyses showed that fragment duplications were the primary factor of amplification in the *TkCPT/CPTL* gene family. Induced by ethylene and methyl jasmonate hormones, the expression levels of most genes increased, with significant increases in the expression levels of *TkCPT5* and *TkCPT6*. Our results provide a theoretical basis for elucidating the role of the *TkCPT*/*CPTL* gene family in the mechanism of natural rubber synthesis and lay a foundation for molecular breeding of *T. kok-saghyz* and candidate genes for regulating natural rubber biosynthesis in the future.

## 1. Introduction

*Taraxacum kok-saghyz* L. Rodin (TKS) is a small, diploid, sexually reproducing perennial plant in the *Asteraceae* family. As early as 1931, it was discovered in valleys of the Tianshan Mountains in southeastern Kazakhstan [1]. *T. kok-saghyz* typically thrives in non-saline or slightly saline floodplain meadows and agricultural canals [2] and can be cultivated in most temperate regions [1]. *Hevea brasiliensis* is the only commercially viable rubber producer that yields high-quality natural rubber (NR) [3]. Nevertheless, due to the susceptibility of *H. brasiliensis* to pathogen infections and its low genetic diversity [4], the exploration of alternative natural rubber resources is imperative. Compared to Brazilian rubber trees, *T. kok-saghyz* has a higher molecular weight of natural rubber [5]. Furthermore, it also has the advantages of a wider planting area, better quality, easier harvesting, and shorter growth cycles, potentially making it a commercially viable alternative to natural rubber as a raw material [6].

Two primary metabolic pathways for the synthesis of isoprenoids in plants have been defined: the methylerythritol (MEP) pathway and the mevalonate (MVA) pathway. Natural rubber is composed of isopentenyl pyrophosphate (IPP)-derived isopentenyl monomers, which are synthesized mainly by the cytoplasmic MVA pathway [7]. The tissues in *T. kok-saghyz* that produce NR are known as laticifers, which contain large quantities of rubber particles (RPs) [8]. Rubber particles are usually spherical or pear-shaped [9], with many proteins attached to the lipid membrane of the rubber particles [10]. This includes proteins, such as cis-prenyltransferase-like (CPTL) protein, small rubber particle protein (SRPP), and rubber elongation factor (REF) [11,12]. Cis-prenyltransferase (CPT), the main component of the rubber transferase (RTase) complex [13], is responsible for the construction of the long hydrocarbon skeleton, which extends the short, full-chain oligopentenyl diphosphate precursor by the sequential addition of the desired number of isopentenyl diphosphate (IPP) molecules which results in the formation of a stretch of cis units [14], which catalyze the elongation of the molecular chain of natural rubber. However, CPTs are incapable of producing high-molecular-mass NR alone [15], and they must interact with other proteins such as CPTL, REF and SRPP to achieve sufficient catalytic activity [16].

CPT, cis-prenyltransferase, also known as *H. brasiliensis* rubber transferase (HRT), is present as prokaryotic CPT in bacteria in the form of undecylenic pentanediol, which functions as a glycosyl carrier in the biosynthesis of peptidoglycan cell walls in bacteria. In plant plastids, prokaryotic CPT is expressed in the form of polyisopentyl glycol, which affects the fluidity of the membrane of vesicles and photosynthesis in plant plastids [17]. The *CPT* gene family has been relatively conservative throughout the course of evolution and can be classified into three categories based on chain length: short chain (C15), medium chain (C50–C55), and long chain (C70–C120) [14]. CPT is divided into two categories based on its enzyme composition: homodimeric enzymes (short-chain and medium-chain enzymes) and heterodimeric enzymes (including long-chain enzymes and rubber synthase) [18]. It has been reported that the gene encoded by *CPT* was first cloned in micrococcus luteus B-P in 1989 [19]. In 2003, Asawatreratanakul isolated and identified two genes, *HRT1* and *HRT2*, encoding cis-isopentenyl transferases from the latex of *H. brasiliensis*. However, expression of recombinant proteins of HRT2 by the prokaryote Escherichia coli revealed that HRT2 did not exhibit rubber-transferring enzyme activity in vitro [20]. In 2011, it was reported that the human CPT formed a heterogeneous protein complex with the Nogo-B receptor and became active in dolichol biosynthesis. In 2012, scientists expressed recombinant proteins of HRT1 and HRT2 in the protoplasts of Saccharomyces cerevisiae and *Arabidopsis thaliana*, exhibiting cis-isopentenyl transferase activity, but did not catalyze the formation of high-molecular-weight natural rubber [18]. The cis-isopentenyl transferase-like enzyme, also known as CPT bridge protein (CPTBP), was shown by Yamashita in 2016 to form a protein complex with CPT, CPTL, and REF, which is involved in the biosynthesis of natural rubber [21]. In the same year, similar complexes consisting of CPT and its binding protein homologues were identified in *Lactuca sativa*, *tomato* and *A. thaliana* [22,23], whereas the dehydropolyphenol bisphosphate synthetases of human NgBR/hCIT and yeast Nus1/Rer2 [24,25] also belong to the group of cis-isoprenyltransferases. CPTL, as a necessary component of the rubber transferase complex, can promote NR biosynthesis by interacting with CPT [26]. Other components of the rubber transferase complex, such as SRPPs, were mainly located on small rubber particles and played a role in stabilizing the structure of rubber particles [27]. REF was also abundantly located on the surface of rubber particles and can improve CPT activity [28].

There have been no studies comprehensively reporting the genome-wide identification and analysis of the *CPT*/*CPTL* gene family in *T. kok-saghyz*, and its specific role is unknown. To systematically investigate the effect of the *TkCPT*/*CPTL* genes on the biosynthesis of *T. kok-saghyz* natural rubber, we have proposed a genome-wide characterization of the *TkCPT*/*CPTL* family members, having analyzed the physicochemical properties, basic structure, and evolutionary relationships of the family members. After treatment with ethylene and methyl jasmonate, members showed different expression patterns. The subcellular localization of some family genes was observed. These results provide new ideas for further studying the effects of exogenous plant hormones on the biosynthesis of natural rubber.

## 2. Results

### 2.1. Identification of the CPT/CPTL Gene Family in T. kok-saghyz

A total of eight *TkCPTs* and two *TkCPTLs* were identified in the TKS genome, which were named TkCPT1~TkCPT8 based on their sequential order in the chromosome. The predicted relevant physicochemical properties are shown in Table 1. The CDS length ranged from 735 bp (TkCPTL1) to 933 bp (TkCPT6), the amino acid residue length ranged from 244aa (TkCPTL1) to 307aa (TkCPT4), the molecular weight ranged from 28.55431 kDa (TkCPTL1) to 35.00990 kDa (TkCPT6), and the isoelectric point prediction ranged from 5.91 (TkCPT1) to 9.32 (TkCPT7). The isoelectricity of TkCPT1 was <7, indicating that it was an acidic protein. The instability index was 31.24 (TkCPT4) to 53.56 (TkCPT7), and the instability indexes of TkCPT3, TkCPT4 and the two TkCPTL members were <40, which indicated that the four proteins were stable. The maximum value of the aliphatic index was 104.39 (TkCPTL2), the minimum value was 80.00 (TkCPT6), and the indices of TkCPT2, TkCPTL1 and TkCPTL2 were all >100. A possible explanation for these results may be that they had better solubility and thermal stability. The results of hydrophilicity analysis showed that the total hydrophilicity indices of the TkCPT/CPTL family proteins all ranged from −0.41 to −0.1; all of them were hydrophilic proteins. The prediction of subcellular localization of the proteins using online software showed that the TkCPT family proteins were localized in the endoplasmic reticulum, whereas TkCPTL1 was localized in the chloroplasts, and TkCPTL2 was localized in the cytoplasm (Table 1).

### 2.2. Secondary and Three-Dimensional Structures Analysis of the TkCPT/CPTLs

The results showed that most of the secondary structures of the proteins of TkCPT/TkCPTLs family members were α-helices, followed by irregularly coiled and β-turns, with the proportion of α-helices ranging from 30.69% (TkCPT2) to 53.41% (TkCPT8), while β-turns ranged from 3.87% (TkCPT6) to 6.90% (TkCPT1), and irregularly coiled ones ranging from 25.62% (TkCPT7) to 39.31% (TkCPT2) varied; the proportion of irregular curls varied from 25.62% (TkCPT7) to 39.31% (TkCPT2) (Table 2). The number of amino acids in each structure accounted for roughly the same percentage, while the secondary structures of each TkCPT/TkCPTL family member were similar to a certain extent, such as TkCPT1, TkCPT2, TkCPT3, and TkCPT4, which were more consistent in terms of their secondary structures.

The three-dimensional structures of eight TkCPT proteins and two TkCPTL proteins were predicted using homology modeling, and the results revealed that, except for TkCPT5 and TkCPT6, there were significant similarities between different members of TkCPT/CPTLs, and thus it could be inferred that these members might have similar protein functions. The three-dimensional structures of TkCPT5 and TkCPT6 were more distinct from the other proteins, but highly similar to each other, possibly due to divergence during evolution (Appendix A).

### 2.3. Evolutionary Relationship, Conserved Structure, Motif Composition, and Gene Structures of TkCPT/CPTLs

Evolutionary analyses showed that TkCPT3 was more closely related evolutionarily to TkCPT4, and TkCPT1 was more closely related to TkCPT2 (Figure 1A). Conserved motif analysis of TkCPT proteins showed that motif 1, motif 2, and motif 3 were present in all TkCPTs (Figure 1B), and in particular, all family members except TkCPT6 contained motif 4. The two members of TkCPTL, on the other hand, had fewer motifs present, only motif 4 and motif 5, but were quite conserved. Analysis of the structural features of TKCPT/CPTL revealed that all proteins contain UPPS structural domains or belong to the UPPS superfamily (Figure 1C). TkCPT family members have different positions and lengths of untranslated regions (UTRs) and coding sequences (CDSs), with the number of CDSs ranging from one to three, and TkCPTL members have more CDSs, totaling six. Except for TkCPT4, members of the same evolutionary branch had similar UTR and CDS features (Figure 1D).

### 2.4. Evolutionary Analysis of TkCPT/CPTL Proteins

To further explore the evolutionary relationship between CPT/CPTL families, a total of six species, including *Helianthus annuus*, *Lactuca sativa* and *T. mongolicum*, which belong to the same family of *Asteraceae*, as well as the model plants *A. thaliana*, *H. brasiliensis* and *Oryza sativa* from the family of *Gramineae*, were compared with *T. kok-saghyz*. Eight CPT members of *T. kok-saghyz*, seven CPT members of *T. mongolicum*, nine CPT members of *A. thaliana*, nine CPT members of *H. annuus*, eleven CPT members of *H. brasiliensis*, seven CPT members of *L. sativa*, three CPT members of *O. sativa*, and a total of ten CPTL members in seven species, respectively, and a total of sixty-four CPT proteins were identified; CPT members showed large differences in numbers across species. These proteins could be divided into six subclades. Proteins in the same subclade may perform similar biological functions. Group IV was the smallest among all the subclades, containing only six proteins, Group I contained seven proteins, Group III contained nine proteins, all CPTL members were present in Group VI and there were no other CPT members, and group V had twenty-one proteins (Figure 2).

### 2.5. Chromosomal Localization of TkCPT/CPTL Genes

In order to better show the evolutionary pattern of the *CPT*/*CPTL* gene family, chromosomal localization of the genomic distribution of the *CPT*/*CPTL* family genes in *T. kok-saghyz* as well as in *T. mongolicum* and *L. sativa* of the same *Asteraceae* family was performed. The TKS genome has a total of eight chromosomes, and members of the *TkCPT*/*CPTL* family were unevenly distributed on four of these chromosomes. All genes were located at the ends of chromosomes and varied in number from two to three: chr1 and chr2 have three, chr4 and chr5 have two. Gene duplication events were present in *TkCPT1* and *TkCPT2*, *TkCPT3* and *TkCPT4*, and *TkCPT5* and *TkCPT6*. The evolutionary relationship subclades were very similar between the species. The *CPT* genes of *T. mongolicum* were all equally dispersed on all four chromosomes, and the genes were present in locations very similar to the distribution of *TkCPT*/*CPTL* on the chromosomes. The *CPT* genes of *L. sativa* were present on different chromosome numbers, with one *LsCPT* present on both chr2 and chr6, two on chr8, and three on chr5 (Figure 3), and the distribution was more dispersed, with most of them near the middle of the chromosomes.

### 2.6. Gene Duplication, and Collinearity Analysis of TkCPT/CPTL Genes

Tandem duplications and segmental duplications were the main causes of gene amplification in plants, and collinearity between species can be used to identify the location of orthologous genes. To explore the amplification and evolutionary mechanisms of the *TkCPT*/*CPTL* gene family, we analyzed the collinearity of the *TkCPT*/*CPTL* gene family. A total of two collinear blocks, *TkCPT3*/*CPT7* and *TkCPTL1*/*CPTL2*, were identified. Homologous genes perform the same function (Figure 4). The prediction of familial gene duplication events revealed that both pairs of genes were caused by segmental duplication. The other members had a variety of duplication types, with *TkCPT1* and *TkCPT2* being tandem duplications and *TkCPT4*, *TkCPT5*, and *TkCPT6* being proximal duplication types. To further analyze the selection pressure between *TkCPT*/*CPTL* genes, we calculated the Ka/Ks ratios of these two gene pairs for non-synonymous substitutions (Ka) and synonymous substitutions (Ks), where the Ka of *TkCPT3*/*CPT7* was 0.16 and the KS was 1.24; the Ka of *TkCPTL1*/*CPTL2* was 0.21 and the KS was 0.83, indicating that both pairs of this family had a Ka/Ks ratio less than one. These data suggested that members of this family had been selected for stronger purifying selection (Table 3).

To further understand the mechanism of gene duplication of *TkCPT*/*CPTLs* and to study the evolution of gene mutation, we constructed a gene duplicator including *T. mongolicum* of the same genus, *L. sativa* and *H. annuus* of the same *Asteraceae* family, *A. thaliana*, which was a model plant, and the rubber plant *H. brasiliensis*. The results showed that there were eleven pairs of collinear genes between *T. kok-saghyz* and *T. mongolicum*, nine pairs of collinear genes with *L. sativa*, three pairs of collinear genes with *A. thaliana*, six pairs of collinear genes with *H. annuus*, and five pairs of collinear genes with *H. brasiliensis*. It is noteworthy that *TkCPT3* and *TkCPT7* had collinear gene pairs in all species (Appendix A), which may have existed earlier in the evolutionary process, and these findings suggest that *TkCPTs* had the highest gene homology with *T. mongolicum* and were more closely related (Figure 5).

### 2.7. Promoter Analysis of TkCPT/CPTL Genes

Promoter cis-acting elements are important transcription initiation binding regions that play an important role in the regulation of gene expression. To understand the possible regulatory mechanisms of *TkCPT* under hormone treatment and abiotic stress, we predicted cis-acting elements in the 2000 bp region upstream of the *TkCPT*/*CPTL* gene using the online database Plant CARE, using a total of 52 cis-acting elements, including TATA-box, CAAT-box, and CCAAT-box (promoter-related); ACE, GT1-motif, and box four (light-responsive element-related); ABRE element (related to abscisic acid); GARE-motif, and TATC-box (gibberellin-responsive element); TGA element (growth-hormone-responsive element); CGTCA-motif (involved in the MeJA reaction) and TGACG-motif; TCA element (involved in the salicylic acid reaction) etc. (Appendix A). All *TkCPT*/*CPTL* genes except *TkCPT1*, *TkCPT5* and *TkCPT8* possessed LTR elements, suggesting that these genes may play a role in cold and drought. These cis-acting elements were classified into five categories, including eleven environmental-stress-related elements, eleven development-related elements, thirteen hormone-responsive elements, eleven light-responsive elements, and six promoter-related elements. Among them, promoter-related and binding-site elements were the most numerous, accounting for 60.66 percent of all cis-acting elements, and light-responsive elements were the least numerous, accounting for only 6.02 per cent. Further analysis of some hormone-related cis-acting elements showed that only *TkCPT3* possesses a P-box, which is related to salicylic acid associated with plant defence against pathogens, and *TkCPT5* and *TkCPT6* possess the CGTCA-motif or TGACG-motif, both of which were responsive to jasmonate (Figure 6), which will provide a basis for the subsequent study of the expression characteristics of *TkCPT*/*CPTL* genes after hormone treatment and function verification.

### 2.8. qRT-PCR Analysis of TkCPT/CPTL Genes

Quantitative real-time fluorescence PCR was used to detect the expression of *TkCPT*/*CPTL* in different tissues. The results showed that the root expression of *TkCPT1*, *TkCPT2*, *TkCPT3*, *TkCPT5* and *TkCPT6* was significantly upregulated after ethylene induction. Among them, the expression of *TkCPT5* and *TkCPT6* was the most significantly upregulated compared with the control, which was presumed to play a key role in the ethylene-regulated pathway, and the addition of exogenous hormones stimulated the expression of these genes. *TkCPT7* and *TkCPT8* were gradually downregulated and basically all *TKCPT*/*CPTLs* were significantly upregulated within 3 h of treatment, followed by a significant downregulation of expression at 6 h. *TkCPT8* was unique, with downregulation of expression at 3 h and upregulation of expression at 6 h, which was different from the expression pattern of other genes, and it was speculated that it might have other biological functions. In leaves, the expression of genes such as *TkCPT1* and *TkCPT5* mostly increased but was less pronounced compared to that of roots, and the experimental results suggest that these genes may be involved in regulatory pathways of ethylene metabolism that regulate growth and developmental processes in specific tissues (Figure 7).

In addition, we treated the roots and leaves of TKS with methyl jasmonate to detect the expression of *TkCPT*/*CPTL* genes. In roots, *TkCPT6*, *TkCPT7*, and *TkCPTL2*, were upregulated at higher levels (Figure 8). The expression of *TkCPT3* gradually increased with treatment time. Some genes reached the highest expression level at 3 h and 6 h equally, whereas in leaves, some genes reached the highest expression level at 3 h and 6 h. The expression of different members varied by tissue. Only *TkCPT8* showed slightly reduced expression levels in roots and leaves and remained at low levels. The results showed that these genes were significantly expressed under methyl jasmonate induction and highly expressed in TKS roots. These experimental results suggested that the *TkCPT*/*CPTL* family was involved in the regulation of jasmonic acid metabolism regulatory pathways.

### 2.9. Subcellular Localization of TkCPT/CPTL

To further explore the functions of TkCPT and TkCPTL proteins, we performed subcellular localization to verify the specific locations where they exert their functions. The TkCPT-pCAMBIA1300-eGFP gene was co-infected with endoplasmic reticulum markers and expressed instantaneously. In the superimposed field, the fluorescence of TkCPT5 and TkCPT6 overlapped within the endoplasmic reticulum appearing as yellow, while TkCPTL1 fluorescence was concentrated on chloroplasts (Figure 9), which was consistent with the predicted results.

## 3. Discussion

Natural rubber is an important high-performance material with unparalleled advantages over synthetic rubber and is widely used in industries such as transportation, medicine and defense [5]. Cis-prenyltransferase (CPT) is a major component of the rubber transferase complex on the surface of rubber particles, catalyzing the extension of the molecular chain of natural rubber [27], whereas cis-prenyltransferase-like enzymes (CPTLs) interact with CPTs to enhance the stability of the CPTs in the ER membranes [29], thus boosting the activity of the CPTs [30]. To thoroughly investigate the function of the *CPT*/*CPTL* family in TKS, this study conducted a genome-wide identification of the *TkCPT*/*CPTL* family based on public genome data, and a total of ten members of the *TkCPT*/*CPTL* family as well as six other species were identified: *H. annuus*, *L. sativa*, *T. mongolicum*, *A. thaliana*, *H. brasiliensis* and *O. sativa*. A total of 64 CPT/CPTL members were divided into six subclades based on their evolutionary relationships (Figure 2). CPT members were distributed in various subclades and were most closely related to *H. annuus* and *L. sativa* of the same *Asteraceae* family. *O. sativa* is a *graminaceous* herbaceous plant, and the number of CPT members was sparse and concentrated. Most members of the rubber tree HbCPT are concentrated in I and III, and among the nine AtCPTs that have been published and identified [14,31], the ACPT (NP.565551) protein sequence contributes to the biosynthesis of long-chain polyisoprenoids [32,33]. LsCPT7 is present on the endoplasmic reticulum in subcellular localizations in tobacco, and the yeast two-hybrid data show that it is closely related to LsCPTL2. The yeast microsomes of LsCPTL2 promote the synthesis of short-cis polyisoprene [34]. In the evolutionary tree, LsCPT7 had evolutionary affinities with TkCPT3 and TkCPT4, and it was speculated that TkCPT3 and TkCPT4 may have an important role in rubber synthesis (Appendix A).

The amino acid sequence of a protein determines how its higher structure is formed and its ultimate function. Multiple sequence comparisons (Appendix A) showed that all TkCPT protein sequences have five characteristically conserved regions, which is the same as previously reported [35], and most of the fully conserved amino acids are involved in catalysis, substrate binding or structural interactions. Some of the important residues, such as Asp (D) in the region I and Asn (W) in region III [36] are highly conserved (100%) and are key amino acid residues related to catalytic activity; Phe (F) (71%) and Ser (S) (71%) in region III, Asp (D) in region IV, Glu (E) in region V [21] and two Arg (R) (100% and 79%) residues [37] are essential for IPP binding. The two members of the TkCPTL do not have the conserved structural domains that are present in both CPT sequences, although some of the bases are also catalytic. The prediction of the protein transmembrane regions showed that all the CPT members did not have a transmembrane helical structure, and interestingly, the two TkCPTL members both have a transmembrane helical region, which might be related to the transport of their special substances or signal transduction (Appendix A). Variation in gene structure is an important feature of family evolution; therefore, we analyzed the basic structure of TkCPT/CPTL and identified a total of ten conserved motifs. TkCPT possesses three highly conserved motifs, which are present in all TkCPTs. Motif 6 and motif 8 are only present in TkCPT3, TkCPT4, TkCPT7. While motif 5 was present in TkCPT3, TkCPT4, TkCPT7 and TkCPT8, which might be related to the unique function of the gene itself (Figure 1B), the rest of the members of the groups had a more similar motif structure. The localization of *TkCPT*/*CPTL* families on chromosomes is more dispersed and more gene duplication events occur, which is more similar to the distribution of *T. mongolicum* on chromosomes, and it is likely that some gene divergence has also occurred during the evolutionary process (Figure 3). Duplicated genes are often subject to some selective pressure to adapt to the external environment. Here, we calculated the Ka/Ks ratio of the two gene pairs of *TkCPT*/*CPTL*, and both gene pairs had Ka/Ks values less than one and underwent purifying selection (Table 3). *TkCPT*/*CPTL* are amplified in a variety of modes, with fragment repeats and tandem repeats being the main modes (Appendix A), which play an important role in the amplification of the number of genes in the family [38,39]. Of course, *CPT*/*CPTL* exercise different functions in different species and environments, and in humans, a subunit of cis prenyltransferase causes a congenital disorder of glycosylation [40]. In the para rubber tree genome, eleven CPT members were determined, but only HbCPT2 protein shows rubber transferase activity [41]. *CPTL*, in addition to its known functions related to rubber synthesis and its homologue in *Arabidopsis* (*LEW1*) were identified in a genetic screen for leaf chlorosis phenotypes. LEW1 mutants are unable to synthesize ethanol efficiently, resulting in defective O- and n-glycosylation of the protein. The results suggest that CPT-like proteins are important in alcohol synthesis in eukaryotes [42].

It has been shown that highly conserved cis-elements are present in many plants and that cis-elements play crucial roles in transcriptional regulatory signaling pathways when plants are subjected to biotic and abiotic stresses. To further elucidate the functions of these genes, we analyzed the promoter regions of the *TkCPT*/*CPTL* family. The cis-acting elements in the promoter region include various types of promoter binding site-related elements (G-box, CCAAT-box, etc.), light-responsive elements (GA-motif, ATCT-motif, etc.), growth and development-related elements (STRE, CAT-box, etc.), and hormone-related responsive elements containing ABA (ABRE), of which *TkCPT1* contains more, which is hypothesized to play an important role in plant resistance to environmental stresses and pathogen infestation. There are also various hormone response elements such as IAA (TGA-element), STRESS (TC-rich repeats, CAT-box), GA (P-box, TATC-box, LTR), JA (TGACG-motif, CGTCA-motif), and SA (TCA-element) (Figure 6), which may play a role in the MYC homeopathic effectors that play important roles in the regulation of gene expression. For example, ABA enhances drought tolerance in plants by binding to MYC cis-acting elements and initiating the expression of downstream genes containing the same kind of elements. The number of ABRE and MYC cis-acting elements in *TkCPT1* is quite high, which may subsequently provide new ideas for the study of key genes involved in the increase in *T. kok-saghyz* rubber production under drought conditions. In a previous report, *CPT* and *CBP* (*CPTL*) genes were found to be downregulated in expression under drought stress by transcriptome analysis of *Parthenium hysterophorus* rubber particles (PR) [43], and it was speculated that this might be related to their involvement in biosynthesis of polyprenol or plastidic polyprenol, which is essential for photosynthesis [44].

Ethylene and MeJA are widely involved in regulating plant growth and development and secondary metabolite synthesis [45], exogenous ethylene promotes seed germination and rooting [46], and MeJA induces the differentiation of laticifer in *H. brasiliensis*, which is an important signal regulating the differentiation and development of laticifer, and exogenous JA enhances the enzyme lipoxygenase in *H. brasiliensis* latex [47]. We treated different tissues of TKS with ethylene and MeJA and performed qRT-PCR to detect the expression levels of *TkCPT*/*CPTL* genes and found that both hormones induced the expression of *TkCPT*/*CPTL* genes in the roots of TKS and that the expression levels of *TkCPT5* and *TkCPT6* were upregulated to a higher level after treatment. In addition, both hormones affect the expression of *TkCPT*/*CPTL* genes in TKS leaves, and overall, MeJA had a greater effect on gene expression in *TkCPT*/*CPTL* leaves than in roots. Interestingly, *TkCPT3* and *TkCPT4* did notcontain MeJA-responsive elements, but the qRT-PCR results showed a corresponding increase in their gene expression. It has been well-documented that jasmonic acid and methyl jasmonate play a key role in natural rubber biosynthesis and response to abiotic stresses in *T. kok-saghyz* [48,49,50]. It has been shown that MeJA can stimulate late secondary metabolism and increase NR content in TKS via the MVA pathway, and that HMGR, FPPS, GGPPS, IDI and ISPF are essential for rubber biosynthesis in the MVA pathway. The transcriptomic data in MEJA-treated TKS roots showed that the expression of these genes was high, and CPT was located downstream of the *GGPPS* gene, which might be related to its increased expression [12,51].

Based on this, we have a more comprehensive understanding of the *TkCPT*/*CPTL* gene families, which provides ideas for subsequent studies to verify the functions of *TkCPT*/*CPTL*.

## 4. Materials and Methods

### 4.1. Plant Materials and Processing Methods

*T. kok-saghyz* (Shihezi City, Xinjiang, China) was selected as the experimental material, cultivated in a growth chamber at 25 ± 2 °Cand 30% humidity, and transplanted in nutrient soil (nutrient soil: perlite: vermiculite = 3:1:1) 15 days after germination. Six-month-old *T. kok-saghyz* was selected as the treatment material, and the roots were immersed into Hoagland’s culture solution of 1 mmol/L methyl jasmonate and 100 μmol/L ethylene. The roots and leaves were collected at the time periods of 0 h, 3 h, 6 h, 12 h, and 24 h, the collected material was immediately placed in liquid nitrogen and stored at −80 °C for preservation, which facilitated the later experimental treatments. The natural rubber in *T. kok-saghyz* was extracted using the alkaline boiling method [52,53].

### 4.2. Identification of TkCPT/CPTL Gene Family and Physicochemical Properties of the TkCPT/CPTL Proteins

The latest genome sequence and annotation files of *T. kok-saghyz* [54] were obtained from the website of the National Centre for Genome Sciences Research (https://ngdc.cncb.ac.cn/, accessed on 10 March 2023), and the identified A.thaliana CPT/CPTL protein sequences were firstly downloaded from the NCBI database as seed sequences, (https://www.ncbi.nlm.nih.gov/, accessed on 10 March 2023), which were screened using local blastp comparisons. Sequences with less than 70% similarity were removed. Secondly, the Hidden Markov Model (HMM) file of the CPT conserved structural domain (PF01255) was downloaded from the PFam database [55] (http://pfam.xfam.org/, accessed on 12 March 2023), and the candidate sequences were submitted to HMM-search. Finally, using Batch CD-Search [56] (https://www.ncbi.nlm.nih.gov/Structure/cdd/wrpsb.cgi/, accessed on 12 March 2023), protein sequences that did not contain complete structural domains were removed to identify TkCPT/CPTL gene family members. The molecular weight and isoelectric point of TkCPT/CPTL were analyzed using Expasy (http://www.expasy.org, accessed on 12 March 2023) [57]. In addition, we used Cell-PLoc 2.0 to predict the subcellular localization of TkCPT/CPTL proteins (http://www.csbio.sjtu.edu.cn/bioinf/Cell-PLoc-2/, accessed on 12 March 2023) [23]. Prediction of the number of transmembrane helices for TkCPT/CPTL proteins were performed using the TMHMM-2.0 web software (https://services.healthtech.dtu.dk/services/TMHMM-2.0/, accessed on 12 March 2023).

### 4.3. Evolutionary Analysis, Gene Structure, Motif, and Conserved Domain Analysis

The construction of evolutionary trees was performed using the maximum likelihood (ML) method with MEGA11 software, version 11.0. To assess the reliability of the tree, a bootstrap value was set to 1000 replicates [58] and applied to calculate the p-distance (Appendix A). Landscaping was performed using the online website EvolView (https://evolgenius.info/evolview, accessed on 20 March 2023) [59]. Motifs of TkCPT/CPTL proteins were analyzed using the online database MEME (https://meme-suite.org/tools/meme, accessed on 20 March 2023), using default parameters with the maximum number of conserved motifs set to 10 [60] (Appendix A). Finally, the gene structure was visualized using Gene Structure View (Advanced) in TBtools v1.099 [61].

### 4.4. Protein Multiple Sequence Comparison, Secondary Structure Prediction, 3D Structure Prediction

The protein sequences of TkCPT/CPTL were compared using Clustal W software v 1.81 and displayed using GeneDoc v 2.7.0.0 software [62]. The secondary structure of TkCPT/CPTL proteins was analyzed using the SOPMA website (https://npsa-pbil.ibcp.fr/cgi-bin/npsa_automat.pl?page=npsa_sopma.html, accessed on 12 March 2023) for analysis. The SWISS-MODEL website (https://swissmodel.expasy.org/interactive, accessed on 12 March 2023) in EXpasy was used to visualize the TkCPT/CPTL protein’s three-dimensional structure [57].

### 4.5. Multi-Species CPT/CPTL Chromosome Localization and Collinearity Analysis

*TkCPT*/*CPTL* gene-related data, including chromosome location data, chromosome length and gene start and end positions was employed. Chromosomal location distribution maps of *CPT*/*CPTL* genes in *T. kok-saghyz*, *T. mongolicum* and *L. sativa* were obtained using the Gene Location Visualize tool from the GTF/GFF module of the TBtools software v1.099. The simple Ka/Ks Calculator, a Ka/Ks calculator in the TBtools software, was used to calculate synonymous substitutions (Ks) and synonymous substitutions (Ka) and the ratio between them, with a divergence time of T = Ks/ (2 × 1.5 × 10^−8^) × 10^−6^ Mya [63]. Comparison of *TkCPT*/*CPTL* collinearity and collinearity with other species was performed using One Step MCScanX-SuperFast in TBtools software v1.099.

### 4.6. Cis-Acting Element Analysis

The sequence of 2000 bp upstream of the start codon of the *TkCPT*/*CPTL* genes was obtained using TBtools software v1.099 and submitted to the PlantCARE website (http://bioinformatics.psb.ugent.be/webtools/plantcare/html/, accessed on 2 April 2023) [64], and the results were imported into the Simple Biosequence Viewer program of the TBtools software v1.099 for visualization. In the HeatMap program of TBtools software v1.099, the number of cis-acting elements and the number of different types of hormone elements were produced, and GraphPad v9.5.1.733 software was used to produce bar charts.

### 4.7. qRT-PCR

Pre-treated plant material was obtained and RNA was extracted using a HiPure HP Plant RNA Mini Kit (Megan, Guangzhou, China) and reverse transcription kit (Vazyme, Nanjing, China) using the cDNA kit for each sample. Each reaction was conducted with three biological replicates. Primers were designed using the primer Premier 5.0 software (https://premierbiosoft.com/, accessed on 25 December 2023) using Tkβ-actin as the internal reference gene, and the reaction procedure was as follows: 95 °C for 15 s, 95 °C for 10 s, 54–62 °C for 20 s, and 72 °C for 20 s for 40 cycles of 95 °C for 5 s, 65 °C for 1 min, and 40 °C for 30 s. The data were quantified by the 2^−∆∆Ct^ calculation method to quantify the relative expression levels of the genes, and the data were statistically analyzed and visualized using GraphPad software 9.5.1. Statistical differences between measurements at different times or with different treatments were analyzed using Duncan’s multiple range test. Differences were considered significant at a probability level of *p* < 0.05.

### 4.8. Subcellular Localization 

To study the transient expression of TkCPT/CPTLs in the leaves of tobacco, the full-length CDS of *TkCPT*/*CPTLs* was PCR-amplified using primers containing KpnI and SalI restriction enzyme (Appendix A) and ligated into the KpnI/SalI vector pCAMBIA1300-eGFP to construct the pCAMBIA1300-35s-eGFP, endoplasmic reticulum localization plasmid tagged with pCAMBIA1300-35S-WAK2-mCherry-HDEL (Puint, Shaanxi, China). The constructed vectors were transformed into *Agrobacterium tumefaciens* GV3101, injected into the bottom of four-week-old tobacco leaves, and incubated in dark culture at 25 °C for 48 h, observed using a laser confocal microscope (Nikon, Tokyo, Japan), with excitation of the GFP. The excitation wavelength of GFP was 488 nm, the excitation wavelength of RFP was 561 nm, and chloroplast auto-fluorescence was 640 nm.

## 5. Conclusions

In this study, we comprehensively identified and characterized the *CPT*/*CPTL* family of *T. kok-saghyz*. We identified eight *CPT* genes (*TkCPT1-TkCPT8*) and two *CPTL* genes (*TkCPTL1-TkCPTL2*) in the *TKS* genome, distributed on four chromosomes. We analyzed their physicochemical properties, gene structures, protein secondary structures, protein three-dimensional structural models, colinear and cis-acting elements in detail. The results showed that all TkCPT genes had upps-related conserved structural domains, Evolutionary tree and covariance analysis provided new perspectives for the subsequent study of the *CPT*/*CPTL* gene family in *TKS*. In addition, the expression patterns of *TkCPT*/*CPTL* genes under the exogenous hormone ethylene and MeJA treatments were also investigated. This study provides important information for the further elucidation of the gene function of the CPT/CPTL family in *TKS* and provides a theoretical basis for subsequent functional validation and study of the regulatory mechanism of natural rubber biosynthesis.

## Figures and Tables

**Figure 1 plants-14-00386-f001:**
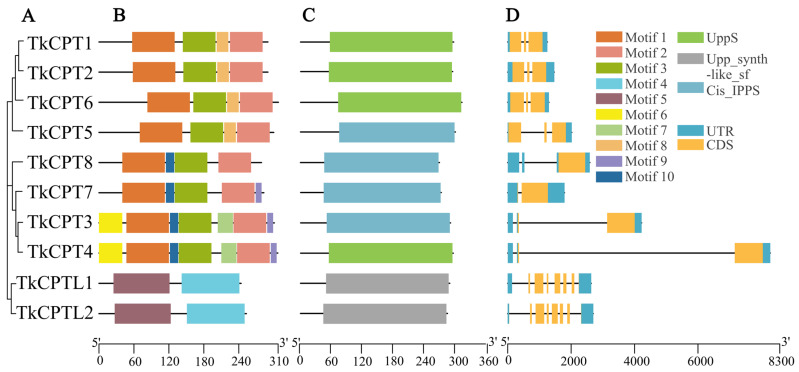
Evolutionary analysis, motif composition, conserved domains, and gene structure of the TkCPT/CPTL family proteins in *T. kok-saghyz*. (**A**) Evolutionary trees based on the TkCPT/CPTL protein sequences was generated with MEGA 11.0 using the Maximum likelihood (ML) method with 1000 bootstrap replications. (**B**) Evolutionary analysis and the served motifs. Ten different kinds of conserved motifs marked with different colors. (**C**) Conserved domains of TkCPT/CPTL proteins, including Upps, Cis_IPPS and Upps-syhth-like-tf. (**D**) Gene structure analysis of TkCPTCPTL. CDS and UTR are represented by yellow and blue boxes.

**Figure 2 plants-14-00386-f002:**
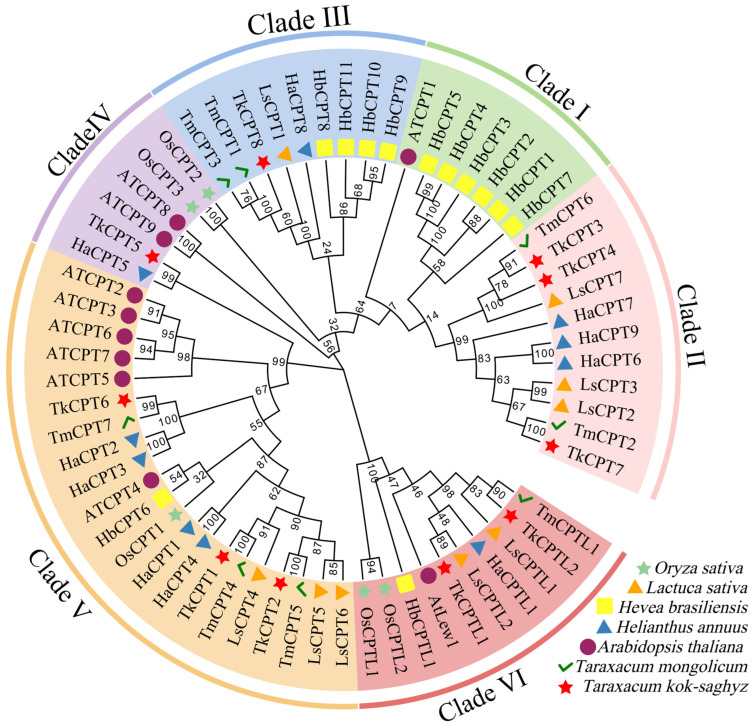
Evolutionary analysis of TkCPT/CPTLs proteins in *T. kok-saghyz*, *T. momgolicum*, *A. thaliana*, *H. annuus*, *H. brasiliensis*, *L. sativa* and *O. sativa*. Evolutionary trees were plotted using the Maximum likelihood (ML) method using the optimal model of T92+G with 1000 bootstrap replicates. Sixty-four proteins were divided into six clades (I–Ⅵ) and identified using different colors. TkCPT/CPTLs proteins from different species were labeled with different shape types. The red pentagram represents *T. kok-saghyz*, the green hook represents *T. momgolicum*, the purple circles represent *A. thaliana*, the blue triangle represents *H. annuus*, the yellow square represents *H. brasiliensis*, the orange triangle represents *L. sativa* and the green pentagram represents *O. sativa*.

**Figure 3 plants-14-00386-f003:**
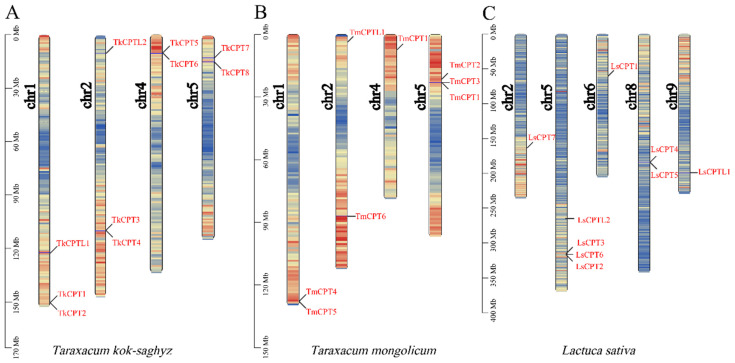
Chromosomal location of *TkCPT*/*CPTL* genes in the *T. kok-saghyz* genome (**A**), *T. mongolicum* genome (**B**) and *L. sativa genome* (**C**). The scale on the left is in megabases (Mb), indicating that the gene chromosome density of the *TkCPT*/*CPTL* gene is indicated in blue to red from low to high, respectively. Chromosome numbers are placed on the left and marked in black. The *TkCPT*/*CPTL* gene numbers are shown on the right side of each chromosome and are marked in red.

**Figure 4 plants-14-00386-f004:**
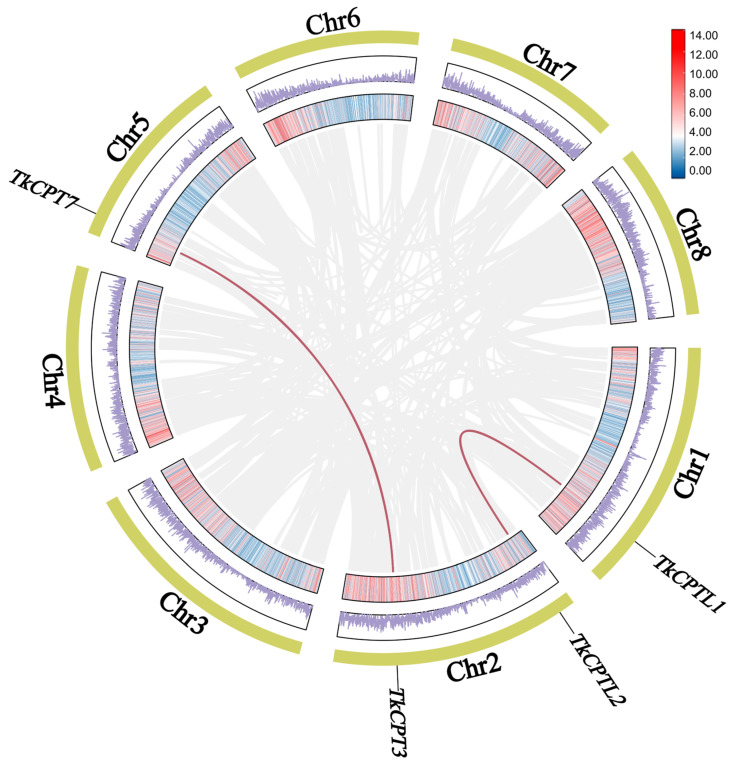
Collinearity analysis of *TkCPT*/*CPTL* gene. The gray lines in the background represent collinear modules in the *T. kok-saghyz* genome. The red lines indicate *TkCPT*/*CPTL* gene pairs with collinearity. The outer yellow rectangle represents the chromosome, the middle rectangle with the purple line represents gene density, and the inner circle represents another form of gene density.

**Figure 5 plants-14-00386-f005:**
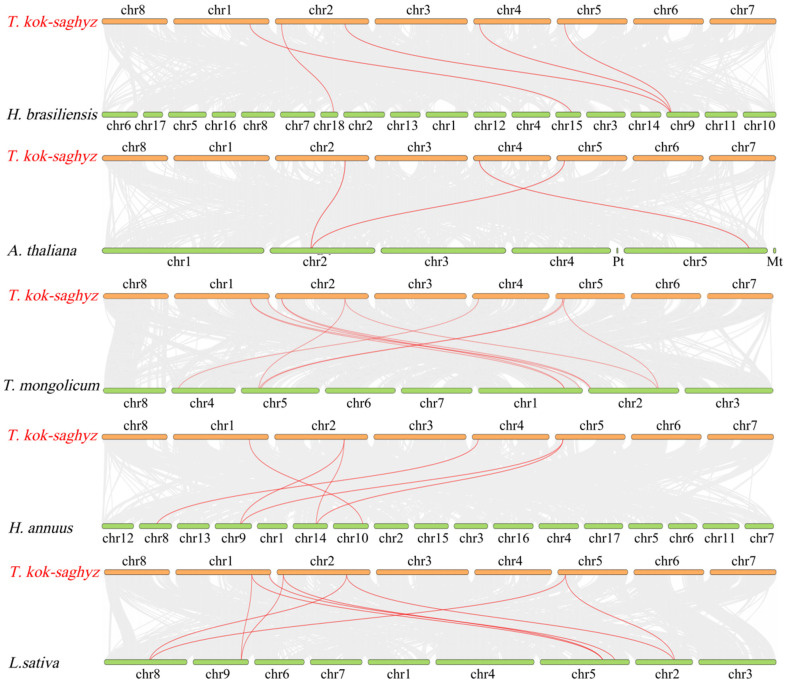
Analysis of collinearity between *TkCPT*/*CPTL* and five other species. The gray lines in the background represent collinear modules in the genomes of *T. kok-saghyz* and other plants. The highlighted red line indicates a collinear *CPT*/*CPTL* gene pair. Species names are placed on the left, *T. kok-saghyz* is indicated in red, and *H. brasiliensis*, *A. thaliana*, *T. momgolicum*, *H. annuus* and *L. sativa*. are indicated in black.

**Figure 6 plants-14-00386-f006:**
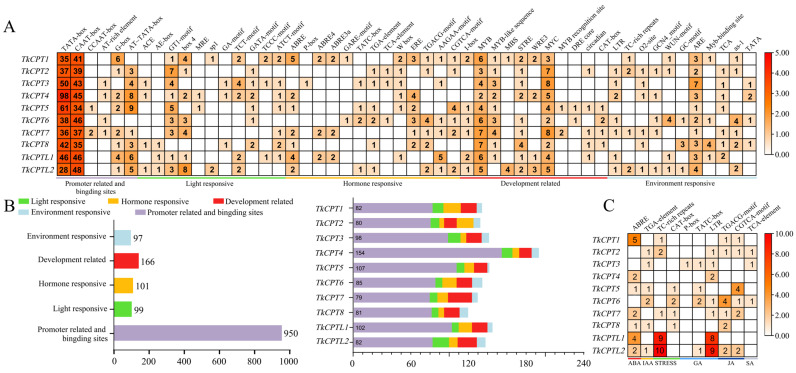
Classification and statistics of cis-acting elements. (**A**) Fifty-two kinds of cis-acting elements were divided into five categories. The different colors on the heatmap represent the number of different cis-acting elements in each *TkCPT* gene. (**B**) Bar chart of the number of cis-acting elements for all genes. (**right**) The different colors on the stacked graph represent the number of each type of cis-acting element in different *TkCPT* genes (**left**). (**C**) Different colors on the stacked graph represent the percentage of different hormone elements in different *TkCPT* genes.

**Figure 7 plants-14-00386-f007:**
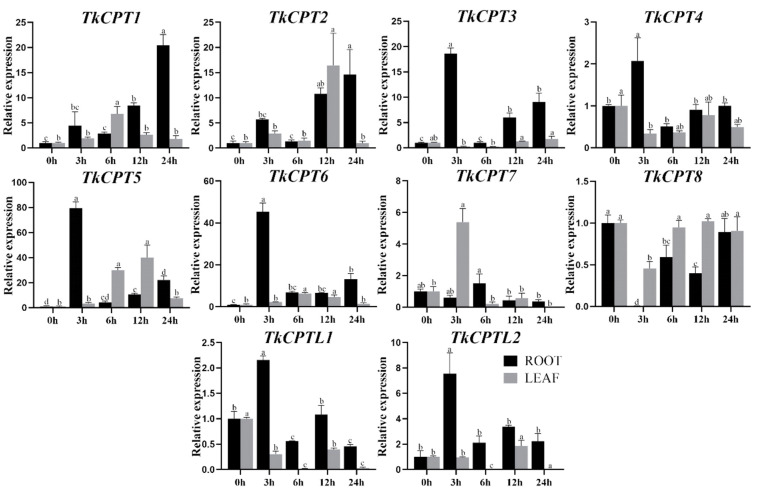
*TkCPT*/*CPTL* expression pattern and physiological indexes under ethylene treatment. Relative expression of *TkCPT*/*CPTL* genes. Data are the average of three independent biological samples ± SD and vertical bars indicate standard deviation. Different letters indicate significant differences at *p* < 0.05, as determined by Duncan’s multiple range tests.

**Figure 8 plants-14-00386-f008:**
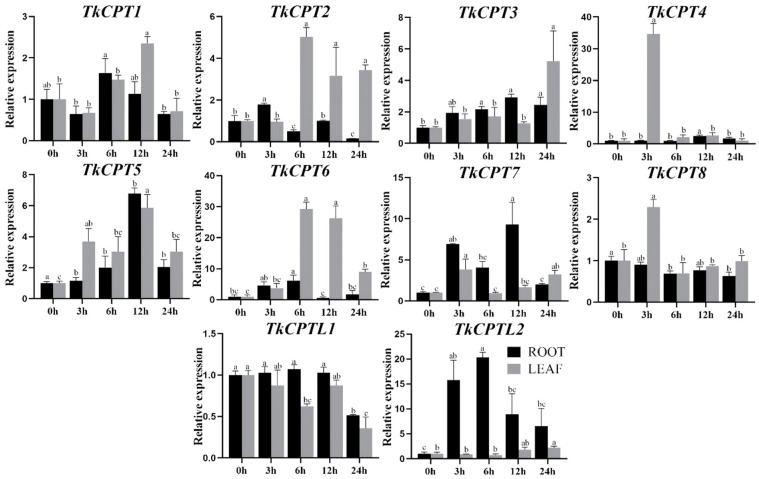
*TkCPT*/*CPTL* expression pattern and physiological indexes under MeJA treatment. Relative expression of *TkCPT*/*CPTL* genes. Data are the average of three independent biological samples ± SD and vertical bars indicate standard deviation. Different letters indicate significant, differences at *p* < 0.05, as determined by Duncan’s multiple range tests.

**Figure 9 plants-14-00386-f009:**
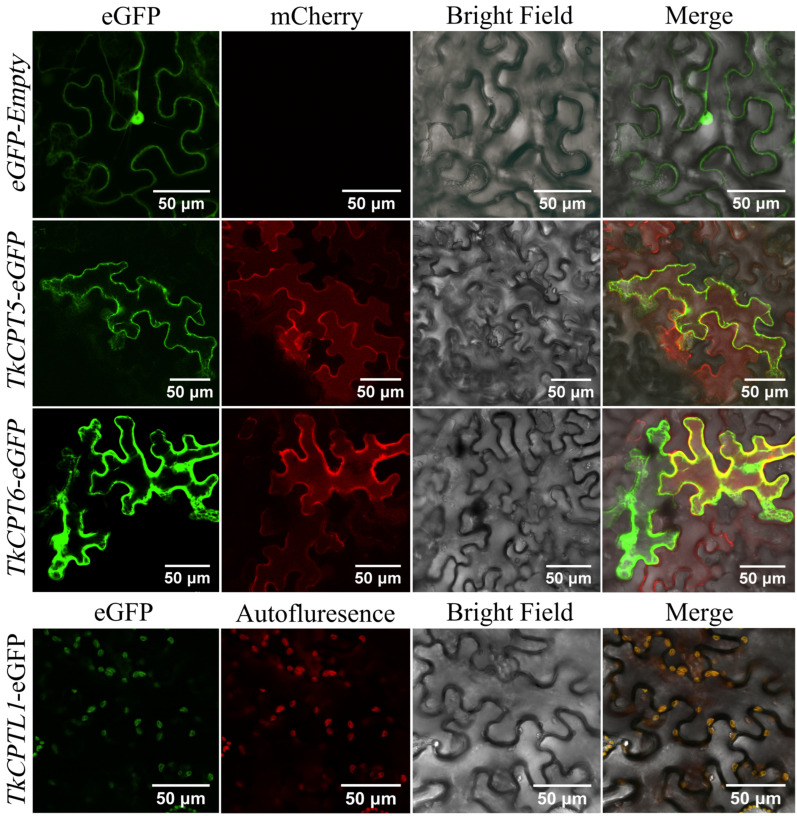
Subcellular localization of TkCPT/CPTL proteins in Nicotiana benthamiana leaves. The empty eGFP vector was used as the control. The endoplasmic reticulum (ER) marker was labeled with mCherry. The targeting signal sequences for the ER (signal peptide of AtWAK2) at the N-terminus and a His-Asp-Glu-Leu (HDEL) motif at the C-terminus.

**Table 1 plants-14-00386-t001:** Detailed information on *CPT*/*CPTL* gene family members in *T. kok-saghyz*.

Gene Name	Gene ID	CDS	aa	MW (kDa)	pI	Instability Index	Aliphatic Index	GRAVY	Subcellular
*TkCPT1*	*TkA01G632490*	873	290	32.61235	5.91	42.63	91.41	−0.196	ER
*TkCPT2*	*TkA01G632500*	873	290	33.10550	6.97	44.29	100.48	−0.215	ER
*TkCPT3*	*TkA02G466190*	906	301	34.28247	8.23	32.90	92.92	−0.148	ER
*TkCPT4*	*TkA02G466270*	924	307	34.85629	9.03	31.24	92.05	−0.171	ER
*TkCPT5*	*TkA04G037510*	903	300	34.25538	7.02	47.05	86.17	−0.281	ER
*TkCPT6*	*TkA04G037560*	933	310	35.00990	8.73	42.53	80.00	−0.409	ER
*TkCPT7*	*TkA05G048860*	846	281	32.73013	9.32	53.56	85.66	−0.342	ER
*TkCPT8*	*TkA05G057980*	840	279	32.01020	8.90	43.74	93.66	−0.164	ER
*TkCPTL1*	*TkA01G518570*	735	244	28.55431	8.29	32.27	103.81	−0.249	Chlo
*TkCPTL2*	*TkA02G042370*	762	253	29.04662	6.40	30.68	104.39	−0.094	Cyto

CDS, coding sequence; aa, amino acid number; MW, molecular weight; pI, isoelectric points; GRAVY, grand average of hydropathicity; ER, endoplasmic reticulum; Chlo, chloroplast; Cyto, cytoplasm.

**Table 2 plants-14-00386-t002:** Predicition of TkCPT/CPTLs protein secondary structure in *T. kok-saghyz*

Protein	Alpha Helix (%)	Beta Turn (%)	Random Coil (%)	Distribution of Secondary Structure Elements
TkCPT1	40.34	6.90	36.90	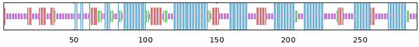
TkCPT2	30.69	6.55	39.31	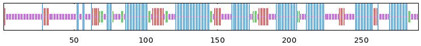
TkCPT3	46.51	5.32	33.55	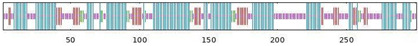
TkCPT4	44.30	5.21	33.22	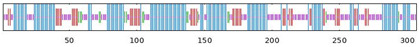
TkCPT5	43.33	6.67	39.00	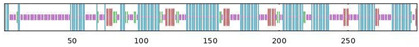
TkCPT6	46.45	3.87	35.48	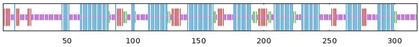
TkCPT7	52.31	6.76	25.62	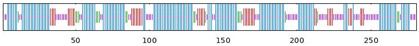
TkCPT8	53.41	5.38	27.60	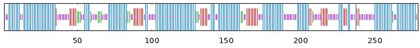
TkCPTL1	48.77	4.10	34.84	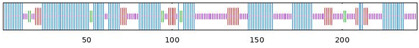
TkCPTL2	53.36	4.74	30.04	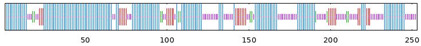

Different colored lines represent different protein secondary structures, blue lines represent alpha helixes, red lines represent extended strand, green lines represent beta turns, and plum red lines represent random coils.

**Table 3 plants-14-00386-t003:** Estimated Ka/Ks ratios and divergence times of the duplicated *TkCPT*/*CPTL* genes.

Gene 1	Gene 2	Non-Synonymous Substitution Rate (Ka)	Synonymous Substitution Rate (Ks)	Selection Strength (Ka/Ks)	Evolution Relationship	Divergence Time (Mya)
*TkCPT3*	*TkCPT7*	0.16	1.24	0.13	Purifying	41.49
*TkCPTL1*	*TkCPTL2*	0.21	0.83	0.25	selection	27.72

## Data Availability

All data generated or analyzed during this study are included in this article.

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
