# Peer review of "Genome-Wide Analysis of the Cis-Prenyltransferase (CPT) Gene Family in Taraxacum kok-saghyz Provides Insights into Its Expression Patterns in Response to Hormonal Treatments"

_plants, 2025, doi:10.3390/plants14030386_

Round 1

Reviewer 1 Report

Comments and Suggestions for Authors

Dear Authors, I have reviewed the manuscript and have the following observations:

The manuscript deals with the species Taraxacum officinale as a natural species for the production of rubber. In the manuscript, a genomic study of the TkCPT/CPTL family has been carried out from this point of view.

The topic of the manuscript fits the journal and I think it is a very novel and topical subject that may be of interest to many researchers. Besides, the topic is topical as synthetic rubber production and application is a very polluting issue and it would be good to use natural materials.

My insights are as follows:

The title, keywords and abstract are appropriate.

Introduction: I suggest deleting the very first paragraph, because the chapter is too long and this part is not specifically related to the topic. Start or end the chapter with a description of the Taraxacum species.

In addition, the chapter does not contain the hypothesis, the research objective - please define it better.

Results, tables and figures are adequate.

The Conclusions section is short and does not show what the hypothesis was, nor the more distant, possibly global, objectives. 

Author Response

The title, keywords and abstract are appropriate.

Introduction: I suggest deleting the very first paragraph, because the chapter is too long and this part is not specifically related to the topic. Start or end the chapter with a description of the Taraxacum species.

We appreciate the thoughtful review and constructive feedback provided by the reviewers. We agree with the reviewers' suggestions and will incorporate the recommended changes into the manuscript.

We have removed the original first paragraph and placed an introduction to T. kok-saghyz in the first paragraph, which adds a section on why T. kok-saghyz was chosen as the gum-producing plant for study. A comparison has been made with the Hevea brasiliensis, which is known to be by far the world's leading gum-producing plant, which highlights the importance of studying T. kok-saghyz in a certain light.

In addition, the chapter does not contain the hypothesis, the research objective - please define it better.

We are very sincerely grateful to the reviewers for their opinion that the target problem of this article is not outstanding. In the last paragraph of the background section of the article, we make a constructive goal to comprehensively report on the genome-wide identification and analysis of the T. kok saghyz CPT/CPTL gene family, with a preliminary understanding of their specific roles. To provide new ideas for studying the effects of exogenous phytohormones on the biosynthesis of natural rubber, as well as to provide further partial insights into the biological functions and molecular mechanisms of rubber substitutes.

Results, tables and figures are adequate.

The Conclusions section is short and does not show what the hypothesis was, nor the more distant, possibly global, objectives.

We are very grateful to the reviewers for carefully reviewing our article and responding with care. In the conclusion section, in addition to the results of this article, we propose constructive conclusions that this article will make certain co-contributions to the subsequent biofunctional studies and molecular breeding of T. kok-saghyz CPT/CPTL, and in a certain sense promote the development of alternative plants for the production of natural rubber.

Finally, we would like to thank the reviewers again for all the above comments, which we have revised one by one and are very grateful.

Reviewer 2 Report

Comments and Suggestions for Authors

The paper performs a complete bioinformatic analysis on a family of genes important for the elongation of a molecule that forms the natural rubber. According to the authors, this may lead to a more efficient characterization of this species as an alternative to Hevea brasiliensis. 

The investigation is quite interesting, but some aspects require major improvement.

Figures 7 and 8: Please include in the figure legend the n number, the meaning of the error bar (error or standard deviation), and the statistical analyses performed to check the significance.

Figure 9: The patterns of CPT5 and 6 seem very different, although in both cases the authors say that the localization is in the ER. The reticular shape can be observed in CPT5, but the nuclear envelope is not clear, while in CPT6 the reticular pattern, typical from the ER, is not obvious. I recommend stacking several pictures from the Z-axis in order to confirm the ER localization by the observation of the reticular pattern. There is a mistake in CPL1a, as according to the lettering, the red stain is mCherry, but in fact they are chloroplasts. Please correct.

There are other points that should be better explained in the text.

The paper indicates that ethylene and MeJA strongly upregulate genes like TkCPT5 and TkCPT6, but some genes like TkCPT3 and TkCPT4 also show increased expression despite lacking MeJA-responsive elements. The authors hypothesize other metabolic regulatory pathways may explain this, but they do not explore these pathways or provide experimental data to validate the hypothesis. Please support this assumption with evidence from the literature or alternative evidence, as in the present form is too speculative.

The duplication analysis indicates purifying selection on TkCPT genes, suggesting functional conservation. However, there is no discussion on whether this conservation limits functional diversity in TKS compared to other rubber-producing species. While purifying selection implies stability, it contrasts with the need for functional specialization to adapt to different environmental conditions. This contradiction is not addressed. Please explain. This could be complemented with what is known about the functional diversity of this family in species like H. brasiliensis.

Author Response

The paper performs a complete bioinformatic analysis on a family of genes important for the elongation of a molecule that forms the natural rubber. According to the authors, this may lead to a more efficient characterization of this species as an alternative to Hevea brasiliensis. 

The investigation is quite interesting, but some aspects require major improvement.

Figures 7 and 8: Please include in the figure legend the n number, the meaning of the error bar (error or standard deviation), and the statistical analyses performed to check the significance.

Thank you for the reviewer's valuable comments. We apologise that this was due to our carelessness and have made the appropriate corrections. In lines 325-327 and 341-343 of the article, we have marked the error bars, symbol meanings, and the significance algorithm used. The results were analysed written through the significance of the correlation.

Data are the average of three independent biological,samples ± SD and vertical bars indicate standard deviation. Different letters indicate significant,differences at p<0.05,as determined by Duncan's multiple range tests.

Figure 9: The patterns of CPT5 and 6 seem very different, although in both cases the authors say that the localization is in the ER. The reticular shape can be observed in CPT5, but the nuclear envelope is not clear, while in CPT6 the reticular pattern, typical from the ER, is not obvious. I recommend stacking several pictures from the Z-axis in order to confirm the ER localization by the observation of the reticular pattern. There is a mistake in CPL1a, as according to the lettering, the red stain is mCherry, but in fact they are chloroplasts. Please correct.

Many thanks to the reviewers for the very valuable comments, for CPT5 subcellular localisation, we used the software to debug, making the picture clearer and more complete, CPT6 we used a new subcellular picture, easy to observe is to negate the inbuilt online located. As well as being very sorry, our labelling of the chloroplasts was the result of carelessness, for which we have rearranged the picture to make it easier to see.

There are other points that should be better explained in the text.

The paper indicates that ethylene and MeJA strongly upregulate genes like TkCPT5 and TkCPT6, but some genes like TkCPT3 and TkCPT4 also show increased expression despite lacking MeJA-responsive elements. The authors hypothesize other metabolic regulatory pathways may explain this, but they do not explore these pathways or provide experimental data to validate the hypothesis. Please support this assumption with evidence from the literature or alternative evidence, as in the present form is too speculative.

Your valuable feedback is greatly appreciated. We have carefully reviewed the literature and included relevant information in the revised manuscript. As the reviewer pointed out, our previous judgement was not based on sufficient evidence.  The TkCPT3 and TkCPT4 gene promoters do not contain cis-acting elements that respond to jasmonic acid signaling in the upstream 2000 bp of the promoter, but there is a significant up-regulation of gene expression after methyl jasmonate treatment.On one hand, the reason is that the cis-acting element that responds to jasmonic acid signaling may be present in the part beyond 2000 bp.On the other hand, it has been well documented that jasmonic acid and methyl jasmonate play a key role in natural rubber biosynthesis and response to abiotic stresses in Rubbergrass. Exogenous methyl jasmonate activates jasmonic acid biosynthesis and signaling, affecting downstream MEP, MVA and natural rubber biosynthetic pathways.

Reference:

Riemann M, Dhakarey R, Hazman M, Miro B, Kohli A, Nick P. Exploring Jasmonates in the Hormonal Network of Drought and Salinity Responses. Front Plant Sci. 2015;6:1077.

Trang Nguyen H, Thi Mai To H, Lebrun M, Bellafiore S, Champion A. Jasmonates-the Master Regulator of Rice Development, Adaptation and Defense. Plants (Basel). 2019;8(9):339.

Dong G, Wang H, Qi J, et al. Transcriptome analysis of Taraxacum kok-saghyz reveals the role of exogenous methyl jasmonate in regulating rubber biosynthesis and drought tolerance. Gene. 2023;867:147346.

The duplication analysis indicates purifying selection on TkCPT genes, suggesting functional conservation. However, there is no discussion on whether this conservation limits functional diversity in TKS compared to other rubber-producing species. While purifying selection implies stability, it contrasts with the need for functional specialization to adapt to different environmental conditions. This contradiction is not addressed. Please explain. This could be complemented with what is known about the functional diversity of this family in species like H. brasiliensis.

We sincerely thank you for your valuable comments, and we have carefully reviewed the literature and added relevant information to the revised draft. In lines 410-418, we added the functions of other plant and animal CPT family members, and the results show that the CPT family has different roles in different environments and species, although some genes have similar functions, which are inseparable from their evolutionary relationships.

Reviewer 3 Report

Comments and Suggestions for Authors

The research topic is trending with the current focus of the scientific community. The authors organize their research framework on gene modification in the Taraxacum kok-saghyz family to produce more pronounced natural rubber in plants. for that purpose analysis of the biosynthesis pathways was performed.

the introduction is very informative pointing out all the key points for successful pathways understanding.  the separate paragraphs describe all important aspects of the first description of polyisoprenes and their polymerization on natural rubber. then a description of the plants and their properties. after that two major metabolic pathways the methylerythritol (MEP) pathway and the mevalonate (MVA) pathway and the importance of  cis-prenyl transferase.

the results are presented logically following the main research premises. the number of picture and tables are appropriate and improve article quality. the discussion is satisfactory and the conclusion summarize all the major findings. 

i have suggestion to authors to clarify FIGURE 4 at the current stage its a little bit fuzzy and could lead to inadequate conclusion.

Author Response

The research topic is trending with the current focus of the scientific community. The authors organize their research framework on gene modification in the Taraxacum kok-saghyz family to produce more pronounced natural rubber in plants. for that purpose analysis of the biosynthesis pathways was performed.

the introduction is very informative pointing out all the key points for successful pathways understanding.  the separate paragraphs describe all important aspects of the first description of polyisoprenes and their polymerization on natural rubber. then a description of the plants and their properties. after that two major metabolic pathways the methylerythritol (MEP) pathway and the mevalonate (MVA) pathway and the importance of  cis-prenyl transferase.

the results are presented logically following the main research premises. the number of picture and tables are appropriate and improve article quality. the discussion is satisfactory and the conclusion summarize all the major findings. 

i have suggestion to authors to clarify FIGURE 4 at the current stage its a little bit fuzzy and could lead to inadequate conclusion.

We sincerely appreciate your affirmation of the content of our article. We have carefully revised the appearance of image 4 to make it clearer and added some content. In line 237241 of the article, we have tried our best to make the description of the results more logical. Thank you again for your good suggestions. Thank you very much.

Round 2

Reviewer 2 Report

Comments and Suggestions for Authors

The paper has been significantly improved.

I can recommend publication.

Author Response

Thank you very much to the reviewers for our revised manuscript recognition, thank you again.